# Microstructure and Mechanical Properties of an Extruded 6005A Al Alloy Composite Reinforced with TiC Nanosized Particles and Strengthened by Precipitation Hardening

**Iria Feijoo [1], Pedro Merino [1] , Gloria Pena [1], Pilar Rey [2] and Marta Cabeza [1,]**

[1]  Materials Engineering, Applied Mechanical and Construction Department, Encomat Group, University of Vigo, EEI, E36310 Vigo, Spain; ifeijoo@uvigo.es (I.F.); pmerino@uvigo.es (P.M.); gpena@uvigo.es (G.P.)

[2]  AIMEN, Technological Centre, Polígono de Cataboi, E36418 Porriño, Pontevedra, Spain; prey@uvigo.es

   Correspondence: mcabeza@uvigo.es; Tel.: +34-986812603

**Abstract:** High-energy ball milling was carried out to disperse 3 vol% TiC nanoparticles (ex situ reinforcement) in a high-pressure inert gas-atomised prealloyed micron-sized 6005A Al alloy (AA6005A), with a Si/Mg atomic ratio of 1.32 powder matrix. Nanocomposite powders were consolidated by hot extrusion in strip shape at 500 °C, followed by a T6 ageing heat treatment. The microstructural features of the consolidated and precipitation hardening nanocomposites specimens were studied using X-ray diffractometry (DRX), differential scanning calorimetry (DSC), scanning electron microscopy (SEM), transmission electron microscopy (TEM), and electron backscatter diffraction (EBSD). The consolidated nanocomposites consisted of approximately equiaxed grains of different grain sizes with a high fraction of high-angle grain boundaries with average misorientation angles of approximately 35°. The nanocomposites showed remarkably higher hardness, Young's modulus, yield, and ultimate strengths at room temperature than the extruded profiles of unreinforced milled AA6005A powders obtained through refinement of the Al alloy grain structure and a strong particle–matrix bonding, although with a drop in their ductility. The consolidated nanocomposite showed a weak response to industrial ageing heat treatment, as demonstrated by microstructural analyses and mechanical tests.

**Keywords:** aluminium alloys; nanocomposites; microstructure; mechanical properties; extrusion; ageing heat treatment

## 1. Introduction

Particle-reinforced metal matrix composites (MMCs) are attractive and well-known materials in which hard and brittle particle reinforcements, usually ceramic, are introduced into a ductile metallic matrix. They exhibit major weight reduction due to the higher strength-to-weight ratio, exceptional dimensional stability, and improved physical and mechanical properties (elastic stiffness and strength, cyclic fatigue characteristics, tribological properties, and creep resistance) compared to unreinforced alloys. They combine the advantages of both the matrix and the reinforcing materials, and they can be adapted to a wide range of combinations by adjusting the size, shape, nature, and volume fraction of the particles [1,2]. Aluminium alloys have been used in most of the particle-reinforced MMCs in the industry because of their low density, good isotropic mechanical properties, excellent corrosion resistance, and reasonable cost [3]. Relatively dilute Al alloys such as 6xxx series Al–Mg–Si alloys are perhaps the most important precipitation-hardening Al base alloys because of their good formability, heat treatability, and easy grain refinement. Among these aluminium alloys, AA 6005A is

a heat-treatable, medium strength, Al–Mg–Si–(Cu) (0.8Si–0.5Mg–0.1Cu) alloy with excellent extrusion performance, because the addition of copper improves their mechanical properties, especially due to ductility, good weldability, and corrosion resistance; it also exhibits a high strength to weight ratio [4–6], so it can be used as a key material in rail transportation and the automotive industry. Despite the indicated characteristics, little research has been done on its use as a metal matrix of composite materials. Ceramic particles have excellent hardness, Young's modulus, strength, chemical resistance, and low thermal expansion [7,8]. Among the various particulate reinforcements used, TiC has proven to be an excellent ceramic reinforcement due to its suitable properties, such as the lower value of density, higher strength, high melting point, low thermal conductivity, superior wear resistance, superior abrasion resistance [7,8], good chemical stability over a wide range of stoichiometry [9], and good corrosion resistance [10].

In recent years, research efforts have been directed towards reducing the size of reinforcing particles to the nanoscale in order to benefit from the interactions between particles and dislocations that, together with the mechanisms known to strengthen MCCs, can lead to a noticeable improvement in mechanical properties [3]. However, the effects of nanosized reinforcements on the mechanical properties are still under research, as well as the strengthening mechanisms working in this type of composite materials.

Several routes have been proposed to produce nanocomposites with the expected properties, mainly based on powder metallurgy (P/M) techniques [10–19] and ultrasonic-assisted casting [20–22], because, although this class of MMCs has been shown to realise high strengths alongside respectable ductility, they are reliant on homogeneous microstructures and are more complicated to process [23]. The most important challenges that P/M techniques face are the high surface area of nanoparticles, which lead to the formation of clusters and inhomogeneous dispersion of the reinforcement, and the poor matrix/reinforcement interface, which results in composites with poor mechanical properties [3]. Particle-reinforced MMCs can be readily shaped by secondary metal working processes, such as rolling and extrusion. Hot extrusion is the most widely used forming process in the manufacture of profiles made of aluminium alloys and their composites for use in very diverse industrial applications. In the extrusion process, large deformations can occur at high temperature and high pressure, leading to substantial changes in the morphology, size and shape of the final grains, microtexture, and the crystallographic parameters, including the orientation and orientation relationships, which in turn are affected by their degree of recrystallisation [24].

The first aim of this work consists of producing and testing AA6005A composites reinforced with ex situ TiC nanoparticles by the P/M route, based on high-energy ball milling (HEBM) and powder consolidation by uniaxial pressed and hot powder extrusion followed by ageing treatment.

According to our previous work [11], it was evident that the presence of TiC nanoparticles had a marked influence on the powder morphology, average particle size, and microstructure of the matrix during the milling process. In addition, a fine homogeneous dispersion of the reinforcement phase into the Al alloy powder was obtained after 10 h of ball milling. Furthermore, the microhardness results of the nanocomposite powder samples showed that their hardness values increased with increased milling time and reinforcement content, and that the contribution of the milling process is greater than that of the reinforcement.

Since the Al alloy used as a matrix of MMC is age-hardenable, it is important to know the effect of the reinforcement on the precipitation behaviour. However, little work has been carried out to verify if the ageing behaviour changes when the reinforcement of heat-treatable Al alloys is reduced to the nanoscale, and if the previously reported effects, such as the acceleration of the ageing kinetics in MMCs, the variation the amount of precipitates according to the change in composition, and the change of precipitation sequences due to the lower presence of quenched in vacancies [25–30], are also valid for metal matrix nanocomposites. Knowles et al. [23] reported that AA6061-based composites with 15 wt % SiC nanoparticles display high strengths after heat treatment, with a hardness increment of 35% owing to the age-hardening effect. However, Casati et al. [31] reported that an AA2618 reinforced

by SiC and oxide nanoparticles showed a weaker response to ageing treatment and exhibited a hardness increment of only 8% with respect to the unreinforced Al alloy, with a slight variation in the activation energies associated with the precipitation process. Furthermore, ageing provides further strengthening to reinforced samples, but also lower elongation to failure. Geng et al. [32] reported that nano-SiC-reinforced AA2024 showed the highest hardness numbers after 4–5 h of thermal treatment at 190 °C, whereas for the unreinforced alloy, the peak aged condition was obtained 12–13 h after solution treatment. Similar results were achieved by Choi et al. [33], who reported that an AA2024 reinforced by carbon nanotubes exhibited its peak hardness after 4 h in an ageing time at 177 °C, where the peak hardness is approximately 1.2-fold higher than the unreinforced alloy, whereas the maximum hardness of the unreinforced alloy is obtained after 18 h. Saheb et al. [34] achieved the opposite results by studying AA6061 reinforced with carbon nanotubes. Nanocomposites indeed exhibited lower hardness values and longer peak ageing times than unreinforced AA6061. There is an inconsistency in the literature, so results and new research efforts should be carried out to better clarify these aspects. Thus, the second aim of this work consists of studying the effect of ball milling and TiC nanoparticles addition on the ageing behaviour to a T6 standard treatment of an extruded AA6005A alloy.

## 2. Experimental Procedure

In the study, aluminium matrix nanocomposite (AlMNC) formed from a 6005A Al alloy (AA6005A) powder reinforced with 3 vol% of TiC nanoparticles (n-TiC) were produced by high-energy ball milling (HEBM). Uniaxial cold pressed and hot powder extrusion (HPE) were used to consolidate the powders.

Prealloyed micron-sized AA6005A powders with a nominal particle size of <63 μm (ECKA Granules Germany GmbH, Früth, Germany), obtained by high-pressure inert gas atomisation to obtain fine and supersaturated microstructures and to increase the limits of solubility, was used as the composite matrix. The reinforcement particles were a nanosized titanium carbide powder (n-TiC) (Ionic Liquid Technologies GmbH, IoLiTec Nanomaterials, Heilbronn, Germany) with a purity of 99% and average particle size of 20 nm. Table 1 shows the chemical composition of the as-received AA6005A (Table 1a) and n-TiC (Table 1b) powders that were determined using elemental chemical analysis, given by the manufacturers. As can be seen, the total Mg + Si amount is 1.49 at % and the Si/Mg atomic ratio is 1.32, which is greater than that of an alloy with a balanced composition (Si/Mg = approximately 1:2, in atomic ratio); therefore, the alloy is designated as an "Si-rich" one. The properties of n-TiC, supplied by the manufacturer, are shown in Table 2.

**Table 1.** (**a**) Chemical composition of the Al–Mg–Si–(Cu) 6005A alloy powders in the as-received condition. (**b**) Chemical composition of the reinforcement n-TiC powders in the as-received condition.

| (a) | | | | | | | | | | | | (b) | | | | | |
|---|---|---|---|---|---|---|---|---|---|---|---|---|---|---|---|---|---|
| Element | Mg | Si | Cu | Cr | Mn | Fe | Zn | Ti | Mn + Cr | Al | Oxygen (ISO 4491/4) | Element | Total C | Fe | N$_2$ | O$_2$ | Ti |
| wt % | 0.57 | 0.88 | 0.11 | 0.01 | 0.13 | 0.18 | 0.11 | 0.10 | 0.14 | Bal. | 0.07 | wt % | 19.91 | 0.20 | 0.30 | 0.85 | Bal. |
| at % | 0.64 | 0.85 | 0.05 | 0.005 | 0.06 | 0.09 | 0.05 | 0.06 | 0.07 | Bal. | 0.118 | at % | 0.999 | 0.00217 | 0.0127 | 0.0322 | 1 |

**Table 2.** Properties of titanium carbide nanoparticles (n-TiC).

| Density (g/cm$^3$) | Melting Point (°C) | Vickers Hardness (GPa) | Elastic Modulus (GPa) | Shear Modulus (GPa) | Thermal Conductivity (w/m/k) | Crystal Structure |
|---|---|---|---|---|---|---|
| 4.93 | 3067 | 24–32 | 400 | 188 | 17–32 | Cubic |

A horizontal attritor ball mill (ZOZ GmbH, model Simoloyer® CM01 ZOZ, Wenden, Germany) with AISI 420 stainless steel balls (5 mm in diameter) and a grinding vessel and rotor, operating with a cyclic mode at 500 rpm for 4 min and 300 rpm for 1 min, with a BPR (ball by powder weight ratio) of 10:1, was used for the HEBM process. These parameters were selected based on previous studies [11,35].

A 3 wt % amide wax (Licowax® C micro powder) was chosen as the PCA (Process Control Agent) to minimise cold welding between Al alloy powder particles, to inhibit agglomeration, and to prevent detrimental welding of the powder to the milling containers and balls during milling. High-purity argon atmosphere was used to avoid excessive metal oxidation during ball milling. The powders were ball milled for 10 h, with stops every 15 min to avoid a significant temperature rise of the powder. The 10 h includes the entire milling time without stops.

Before HPE, the AA6005A-3 vol % n-TiC milled composite (AlMNC), and the unreinforced milled Al alloy (AA6005A) powders were degassed in a furnace at 500 °C for 1 h under an argon atmosphere, uniaxially cold pressed at 200 MPa into cylindrical Al containers, and finally, were pre-heated at 500 °C for 1 h inside the extrusion container to achieve the extrusion temperature in the whole material. Powder samples in the cylindrical container (40 mm diameter and 100 mm length) were extruded at 500 °C, at a ram speed of 0.5 mm/s and an extrusion ratio (ER) of 7:1, which produced a true strain of 1.9. A plane die was used to obtain rectangular section profiles (40 × 3 mm) and 1000 mm length. At the exit of the die, the profiles were quickly cooled (air cooled) at room temperature (RT) to avoid the nucleation and growth of incoherent Mg–Si phases during cooling. Then, T6 heat treatment, was performed on the extruded profiles of AA6005A-3 vol % n-TiC. T6 heat treatment involved a solid-solution treatment (SST) at 530 °C for 1 h to obtain a complete solution of Mg, Si, and Cu atoms, quenching into iced water, and immediately afterwards, an ageing treatment at 175 °C for 8 h, followed by air-cooling. A temperature of 175 °C was chosen for the isothermal heat treatment because it is commonly used in the industry and will cause hardening phases to precipitate.

The X-ray diffractometry technique was employed for phase analysis using a Siemens model D-5000 diffractometer, with Cu Kα radiation (λ = 1.54056 Å). The source was operating under an accelerating voltage of 40 kV with a tube current of 30 mA. The scanning range was 2θ = 30–90°, with a step size of 0.02° and a counting time of 10 s per step. The peaks were identified using EVA software (Bruker AXS Inc., Madison, WI, USA) and compared with the standard values from the International Centre for Diffraction Data's Powder Diffraction File (ICDD-PDF).

Differential scanning calorimetry (DSC) was carried out on extruded profiles to identify the precipitation reactions on a Mettler Toledo DSC 822 instrument. All DSC scans were realised from room temperature to 600 °C. Specimens were submitted to DSC under a hydrogen atmosphere (50 mL/min) with scanning rates of 20 °C/min or 40 °C/min. Platinum pans were used, and high-purity annealed aluminium (99.99%) was the reference material. In order to obtain a baseline (which depends on, among other factors, the different heat capacities of the reference and sample pans), two runs were carried out. The first run was conducted using the high-purity Al discs of approximately the same mass in both pans. Then, the second run was carried out after replacing the Al disc in the sample pan by a disc of the material under study. Subtraction of the heat flow in the first run from the heat flow in the second run results in a signal that allows for the calculation of the heat flow due to reactions in the sample, as described elsewhere [36]. Specimens for DSC were subjected to SST for 1 h at 530 °C in an argon atmosphere and then quenched into iced water. Heating in the DSC commenced within 10 min of quenching in order to avoid natural ageing until the DSC experiments were carried out. Repeated DSC measurements at the different scan rates were made to ensure reproducibility. The utility program provided with the thermal analyser was used in calculating the specific heat capacity data, enthalpies of reaction for each phase transformation, and the amount of each phase transformed with respect to temperature.

Microstructural characterisation was performed in the as-extruded and aged conditions from the intermediate sections of the strips. Samples were sectioned in the transverse plane to the extrusion direction. For submicron observations and STEM analysis, we used a 200 kV field-emission transmission electron microscope (JEOL JEM-2010F, Tokyo, Japan) equipped with a scanning unit and an EDS detector. Thin foils for TEM were prepared by using a Helios NanoLab 600 Focused Ion Beam (FIB) and the in-situ lift-out method.

Grain sizes and grain boundary characteristics of the AA6005A matrix were assessed in the cross-section of extruded profiles using the electron backscatter diffraction (EBSD) method. The specimens were mechanically ground and polished using diamond pastes and a fine colloidal silica suspension for a long time. The acquisition of EBSD data was done using a 3D total analysis (Helios NanoLab 400 FIB-SEM, FEI, Thermo Fisher Scientific, Waltham, MA, USA) equipped with a high-resolution EBSD detector (7000F JEOL, Tokyo, Japan) operating at about 20 mm working distance. Orientation maps were created on a rectangular grid with a step size of 100 nm from a representative area. The corresponding data acquisition and processing were carried out using the Channel 5 software (Oxford Instruments HKL, Abingdon, UK). Misorientation angles between adjacent grains ($\theta$) of $2° < \theta < 10°$ were considered low-angle grain boundaries (LAGBs), whereas $\theta > 10°$ were considered random high-angle grain boundaries (HAGBs). LAGBs and HAGBs are shown in EBSD maps in white and black lines, respectively. Different clean-up options were employed to eliminate the points that were not indexed correctly. Since the orientation uncertainty or orientation noise in ball milling composites is significantly greater than in the rest of the materials, $\theta$ below $2°$ were not measured in order to avoid spurious boundaries. Moreover, a minimum grain size of 100 nm and minimum confidence index (CI) of 0.1 were applied. These limits were used for all samples. Grain size distribution was calculated by EBSD as well. Grain size distribution was obtained from the orientation maps by the software reconstructions of grains, omitting LAGBs and grains that intersect with the edge of the maps.

After microstructural characterisation, the mechanical properties were evaluated. Vickers microhardness ($\mu$HV) measurements were found for each sample using an EMCOTEST DuraScan micro hardness tester (EMCO-TEST, Kuchl, Austria) according to UNE-EN-ISO 6507-1:2006, using a load of 100 gf and a dwell time of 15 s. The values recorded for each sample were made from an average of 10 indentations at a plane perpendicular to the extrusion direction of the polished samples. Tensile stress–strain testing was conducted at room temperature using a 250 kN MTS tensile testing machine under an engineering strain rate of $8.3 \times 10^{-4}$ s$^{-1}$, along the extrusion direction. A MTS 25 mm extensometer was used to obtain the axial strain during loading. The yield stress was calculated on a 0.2% offset. Tensile test specimens with a cross-section of $3 \times 6$ mm$^2$, gauge length of $30.36 \pm 0.15$ mm, and total length of 105 mm were prepared according to UNE-EN ISO 6892-1:2017 Standard Appendix D, small-size specimen. All the specimens were tensioned until fracture.

## 3. Results and Discussion

### 3.1. Microstructural Characterisation of the Milled Powders

The morphology and shape of the as-received micron-sized AA6005A reinforced with 3 vol % of 20–30 nm diameter n-TiC particles and ball milled powders were depicted in detail in our previous work [11]. Starting with, for the most part, a spherical shape, the Al alloy powder particles underwent morphological modifications promoted by the repetitive deformation, fracture, and welding phenomena that occurred during ball milling. After being milled for 10 h, the powders appeared as irregular agglomerates of smaller particles. In addition, a fine homogeneous dispersion of the reinforcement phase into the Al alloy powder was obtained after ball milling. The evolution of the morphology and shape of the AlMNC powders with the milling time was different to that observed in the AA6005A powders, indicating that the HEBM is affected by the presence of hard reinforcement nanoparticles.

Figure 1 shows the X-ray diffraction (XRD) spectra, obtained on longitudinal sections along the extrusion direction of profiles of unreinforced A6005A alloy, AlMNC, and AlMNC after T6 heat treatment of milled powder. An XRD spectra of the aluminium matrix alloy after HEBM for 10 h were studied in our previous work [37], and in the spectra from the AlMNC samples in the as-extruded and aged conditions TiC phase is clearly visible, together with the aluminium matrix alloy.

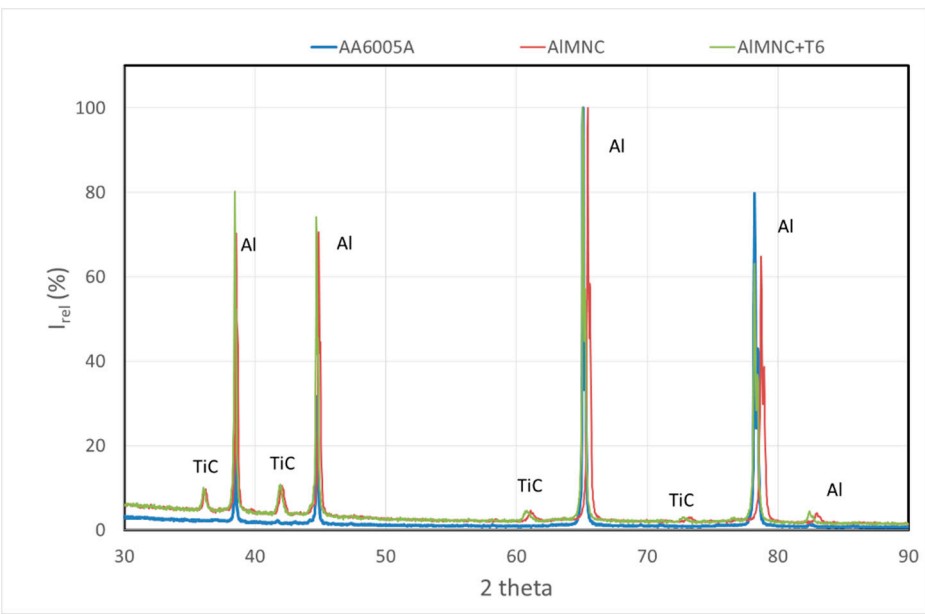

**Figure 1.** X-ray diffraction (XRD) patterns obtained on longitudinal sections of extruded profiles of unreinforced A6005A alloy, aluminium matrix nanocomposite (AlMNC), and AlMNC after T6 heat treatment of milled powder.

There are not high levels of secondary precipitation after T6 treatment in the AlMNC because no second hardening phase is detected by X-ray diffraction.

Figures 2–4 show STEM images representative of the extruded profiles of unreinforced AA6005A and AlMNCs high-energy milled powders. Figure 2a shows a STEM image of the unreinforced AA6005A alloy in the as-extruded condition. A fine-grained structure with a wide size distribution is visible. White <150 nm isolated particles with a well-defined shape within the AA6005A alloy and at the grain boundaries are shown by white and red arrows, respectively. EDS elemental mapping analysis was carried out on these second phases, and the results are displayed in Figure 2b, which suggests that they are Al–Fe–Si–Mn intermetallic particles. Local areas with an accumulation of irregular flakes at the grain boundaries, as shown by blue arrows, can also be seen. Figure 2c shows images at higher amplification of an area of accumulation of these irregular flakes. EDS elemental mapping analysis carried out on this area strongly suggests that these flakes are Mg–Al oxide, as a consequence of ball milling, because during the blending process, new AA6005A matrix surfaces are continuously formed, due to repeated fracturing. In this situation, oxygen may react with Al to form alumina dispersoids and/or with Mg to form Mg–Al oxide [38].

A combination of <150 nm n-TiC agglomerate particles together with particles of <150 nm isolated precipitates in a matrix of aluminium alloy similar to that observed in Figure 2 can be displayed by an EDS elemental mapping analysis for the AlMNC milled powders profiles in the as-extruded condition (Figure 3). As already demonstrated in the our previous work [11], HEBM resulted in a uniform dispersion and incorporation of the n-TiC particles reinforcement in the Al alloy powder matrix, with a small fraction forming clusters with an average size around 150–200 nm. Therefore, the n-TiC nano-agglomerate particles are maintained after the extrusion process.

The microstructure of the extruded profiles of AlMNC milled powders in the aged condition is shown in Figure 4. It has been found to have a grain size <1 μm and an even dispersion of <150 nm n-TiC agglomerated particles within the Al alloy matrix. In addition, it contains another intermetallic precipitated inside the Al grains and at the Al grain boundaries, such as Al–Fe–Mn–Si intermetallic particle with size <1 μm from the Al matrix alloy. During STEM analysis, no clear aluminium carbides or intermetallics were observed at the Al alloy matrix–TiC particle interface, agreeing with the XRD findings. As the TiC has a strong interfacial bond with Al, it can yield $Al_4C_3$ and $Al_3Ti$ at the interface

by a chemical reaction between TiC and the Al matrix during long high-temperature treatments during the composite fabrication [39]. In the present work, the whole process was performed in the solid state, and the temperature and time of ageing heat treatment were not sufficient to produce the formation of any carbide or intermetallics at the Al matrix–TiC particle interface.

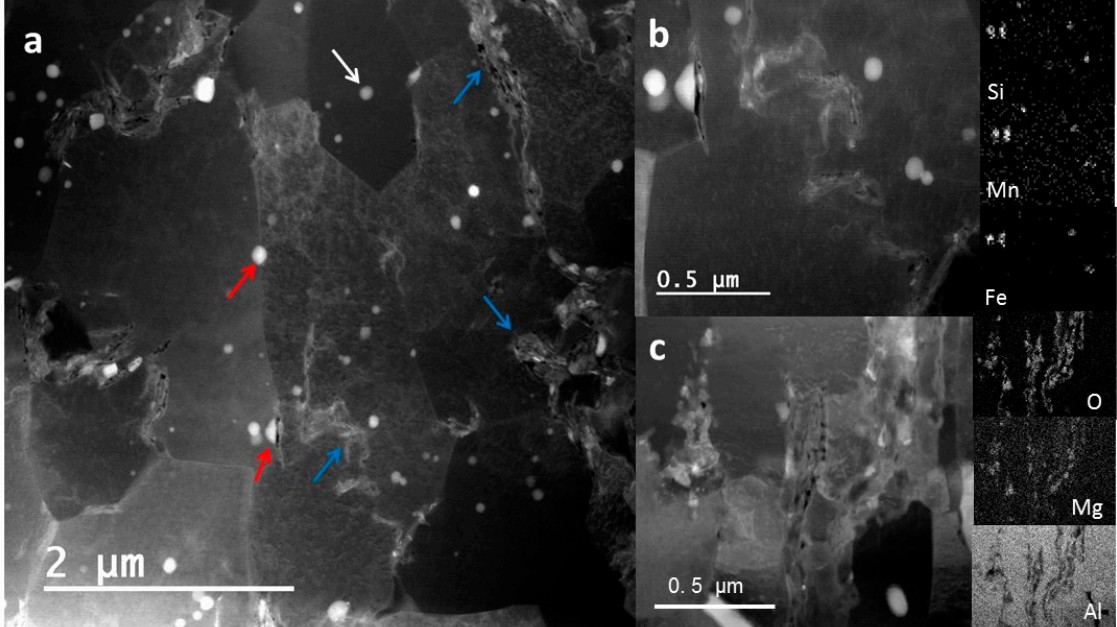

**Figure 2.** (**a**) STEM image of the microstructure of AA6005A profiles (white arrows show isolated precipitates within the Al matrix alloy, red arrows show isolated precipitates at the grain boundary, and blue arrows show local areas with an accumulation of irregular flakes). (**b**) STEM images corresponding to the precipitates at the white and red arrows and EDS elemental mapping analysis results on these precipitates particle. (**c**) High-amplification images of an area of accumulation of irregular flakes, and EDS elemental mapping analysis results on irregular flakes.

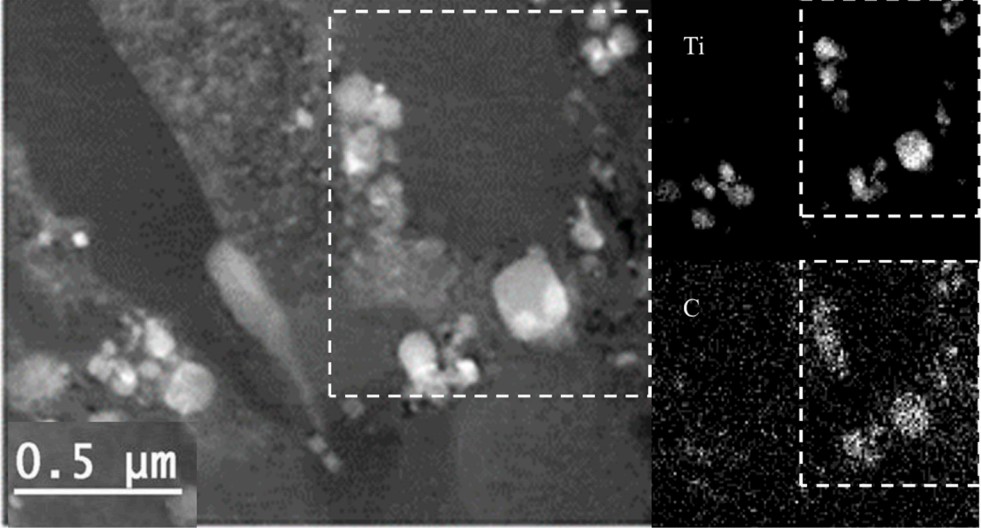

**Figure 3.** STEM image of the microstructure of AlMNC profiles and EDS elemental mapping analysis.

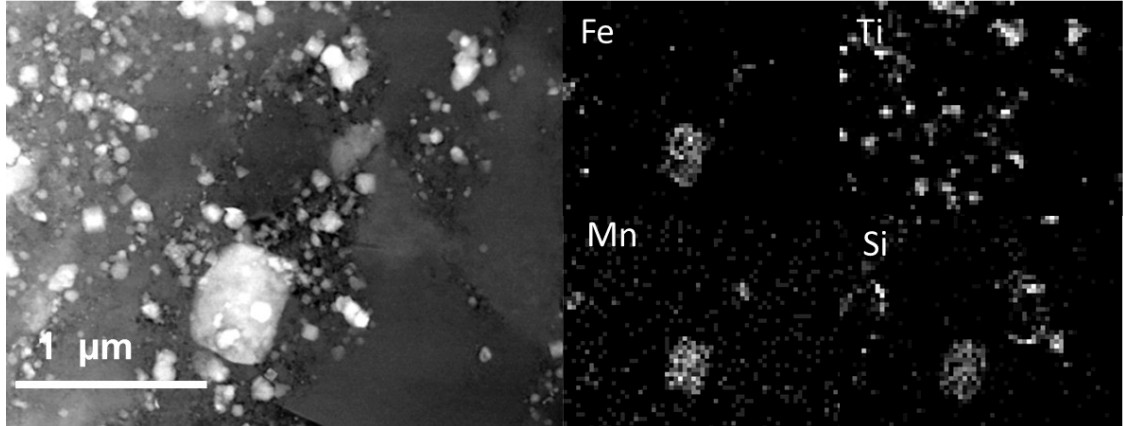

**Figure 4.** STEM image of the microstructure of AlMNC-T6 profiles and EDS elemental mapping analysis.

There is not evidence in the STEM studies of precipitation hardening during T6 treatment. DSC studies try to clarify this.

The extruded profiles of unreinforced and unmilled alloy (UM-AA6005A) and high-energy milled of AA6005A and AlMNC were studied. Figure 5 shows the DSC thermograms from all the samples after SST. DSC scans show the following characteristics that were correlated with the precipitation processes. It is clear that the HEBM process produces changes in the sequence of precipitation from SST for this alloy.

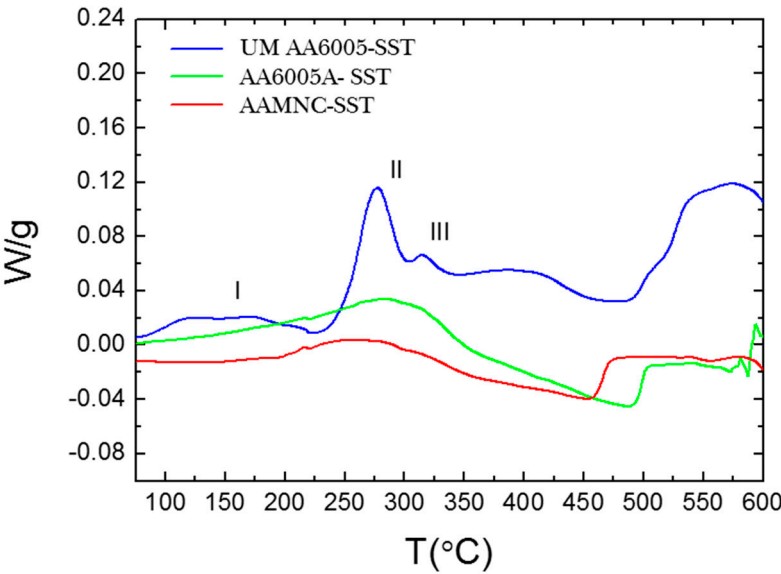

**Figure 5.** Differential scanning calorimetry (DSC) curves of the UM-AA6005A, AA6005A, and AlMNC extruded profiles after solid-solution treatment (SST).

The first peak (I), at approximately 100 °C, is assigned to GP (Guinier-Preston) zones and/or mainly Si clusters formation [36]. The formation of GP zones is weak in the AA6005A and undetectable in the AlMNC. The weakness of the formation of GP zones has been attributed to the low presence of quenched-in vacancies due to the HEBM process [31]. In the case of reinforced materials, the low vacancy density is still lower because the higher dislocation density expected in composites compared with unreinforced alloys leads to the annihilation of the excess of vacancies in the reinforced materials [36], restraining the GP precipitation, because it is known that the vacancies are preferred sites for the heterogeneous nucleation of GP zones [36]. In any case, a prior precipitation of stable GP zones during quenching may have occurred [27]. There is an overlap in the metastable precipitate phases β″ (peak III)

and β′ (peak II) in HEBM materials, whereas in the AA6005A, this overlap is only partial. Two partially overlapping peaks, β″and β´, are also apparent in the investigations performed by other researchers on Al–Mg–Si alloys [36,40,41]. The temperature intervals of the overlapped (β″, β´) precipitates in the DSC thermograms are similar to results obtained by Alberto Borrego et al. [36] in similar materials upon similar processing (extrusion) conditions at the same heating rate. This observation also agrees with the works of Dutta and Allen [40] and Edwards et al. [41], although the temperature intervals are shifted to higher temperatures in the present work. This is most likely due to the faster heating rate that was used in the present investigation (20 °C/min, cf. 10 °C/min for Dutta and Allen and 5 °C/min for Edwards et al.), because in thermally activated processes, the heat effects shift to higher temperatures with an increasing heating rate [27].

### 3.2. EBSD Analysis

The grain size of the Al alloy matrix and microstructural details of the extruded profiles from the mechanically milled powders of AA6005A, AlMNC, and AlMNC after T6 heat treatment, as obtained by EBSD analysis, are presented in Figures 6–8, and Table 3.

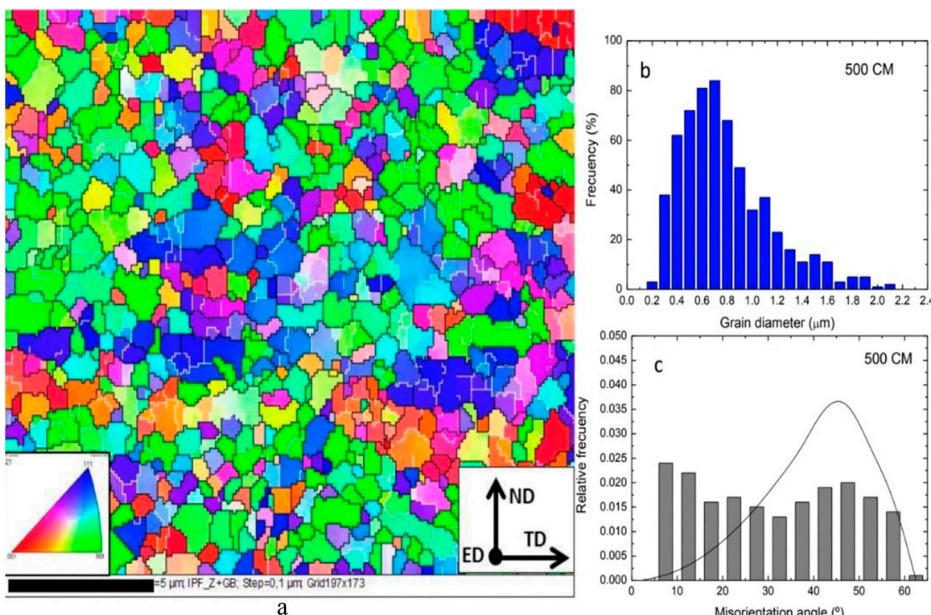

**Figure 6.** (**a**) Electron backscatter diffraction (EBSD) Crystal Orientation Map (COM) with inverse pole key for cubic materials inset in the bottom left corner, (**b**) grain size distribution histogram and (**c**) misorientation angle distribution histogram obtained from cross-sections of aluminium matrix of extruded profiles of unreinforced AA6005A-HEBM (high-energy ball milling) powder. Different colours reflect the crystal lattice orientation positions in the Euler space. Low-angle grain boundaries (LAGBs) and high-angle grain boundaries (HAGBs) are denoted by white and black lines, respectively.

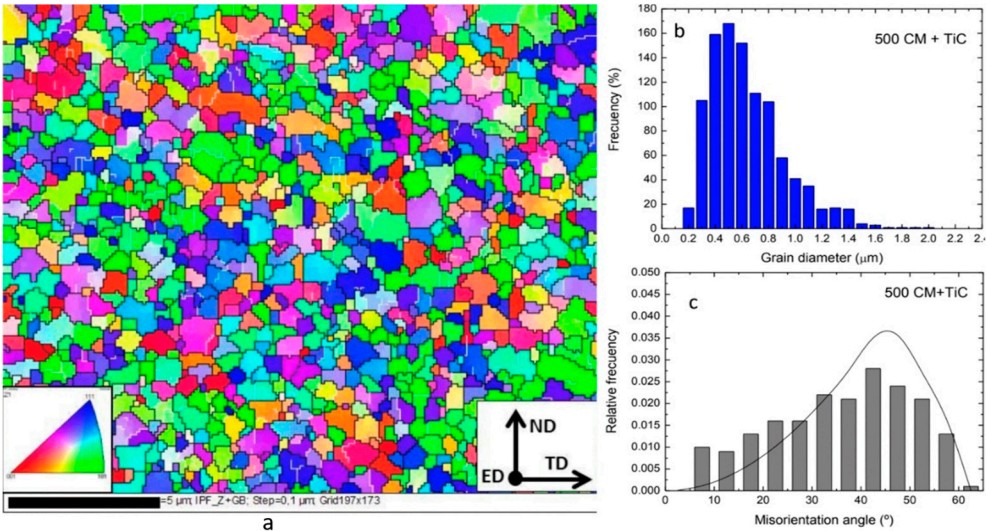

**Figure 7.** (**a**) EBSD Crystal Orientation Map (COM) with inverse pole key for cubic materials inset in the bottom left corner, (**b**) grain size distribution histogram, and (**c**) misorientation angle distribution histogram obtained from cross-sections of aluminium matrix of AlMNC-HEBM in the as-extruded condition. Different colours reflect the crystal lattice orientation positions in the Euler space. LAGBs and HAGBs are denoted by white and black lines, respectively.

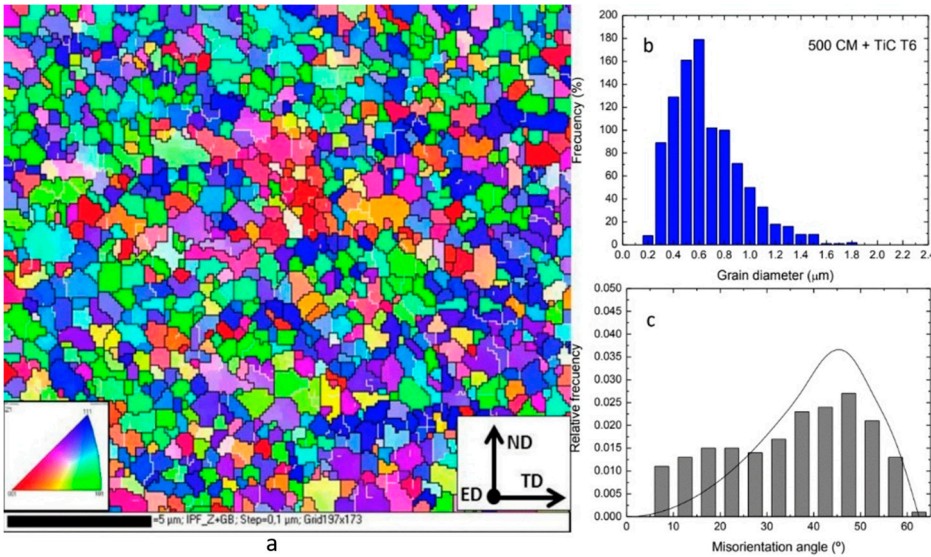

**Figure 8.** (**a**) EBSD Crystal Orientation Map (COM) with inverse pole key for cubic materials inset in the bottom left corner, (**b**) grain size distribution histogram, and (**c**) misorientation angle distribution histogram obtained from cross-sections of the aluminium matrix of the extruded profile of AlMNC-HEBM in the aged condition. Different colours reflect the crystal lattice orientation positions in the Euler space. LAGBs and HAGBs are denoted by white and black lines, respectively.

**Table 3.** Boundary misorientation distributions in extruded profiles of AA6005A-HEBM and AlMNC HEBM, before and after T6 heat treatment powders obtained from EBSD analysis. HPE: hot powder extrusion.

| Misorientation Study | AA6005A HEBM-HPE | AlMNC HEBM-HPE | AlMNC (T6) HEBM-HPE |
|---|---|---|---|
| Fraction of low-angle grain boundaries (LAGBs) (%) | 23.70 | 9.80 | 12.37 |
| Fraction of high-angle grain boundaries (HAGBs) (%) | 76.30 | 90.20 | 87.63 |
| Average misorientation angle (°) | 31.24 | 35.79 | 35.35 |

Figure 6 shows the EBSD results of the extruded profiles of AA6005A-HEBM. As seen in Figure 6a, the microstructure consisted of a wide size distribution of different grain sizes ranging from 160 nm to 2.3 μm with an average grain size of 745 nm and dispersion variance, $D^2X$, of 0.137. Approximately 79% of the grains have a size ≤1 μm, as shown in the histogram of grain size distribution (Figure 6b).

The histogram of misorientation angle distribution in Figure 6c is close to the random MacKenzie distribution curve (a superimposed theoretical distribution for an aggregate of randomly oriented grains) that reveals a random microstructure with a low fraction of deformed grains. The HAGBs are abundantly present in the entire microstructure (approximately 76%) with an average misorientation angle of approximately 31° (Table 3). This confirms the weak microtexture of the extruded strip samples due to no initial texture prior to extrusion, since they were elaborated by P/M, the small true strain imposed on the material, and the dynamic recrystallisation during the HPE process [42–44].

Figure 7 shows the EBSD results of the extruded profiles of AlMNC-HEBM. As seen in Figure 7a, the microstructure also consisted of different grain sizes, from a minimum of 113 nm to a maximum of 1.929 μm with an average grain size of 609 nm and a dispersion variance, $D^2X$, of 0.08. Approximately 86.5% of the grains have a size ≤ 1μm, as shown in the histogram of grain size distribution (Figure 7b), which indicates that a microstructure with an ultrafine grain (UFG) size has been obtained. It is evident that for the same processing conditions, the grain size is smaller and more uniform that in the unreinforced monolithic matrix; this indicates that the main role of n-TiC particulate size and distribution is to modify the matrix grain size. The histogram of misorientation angle distribution in Figure 7c is closer to the random MacKenzie distribution curve than that of the extruded profiles of AA6005A HEBM, which reveals a more random microstructure with a slight increase in the average misorientation angle, as seen in Table 3.

Figure 8 shows the EBSD results of the extruded profiles of AlMNC-HEBM after T6 heat treatment. As seen in Figure 8a, the microstructure also had a UFG size, from a minimum of 113 nm to a maximum of 1.730 μm with an average grain size of 592 nm and a dispersion variance, $D^2X$, of 0.07. Approximately 91% of the grains have a size ≤1 μm, as shown in the histogram of grain size distribution (Figure 8b), and there has been perceptible grain refinement. The histogram of misorientation angle distribution in Figure 8c is closer to the random MacKenzie distribution curve than that the extruded profiles of AlMNC-HEBM without T6 heat treatment, which reveals a more random texture. A higher random HAGBs fraction (approximately 90%) with an average misorientation angle of approximately 36° was noticed in the AlMNC-HEBM than in the AA6005A-HEBM in extruded conditions, which reveals that the small true strain during the HPE process was not been sufficiently modified by the presence of n-TiC reinforcing particles to obtain a strong microtexture. After a T6 heat treatment was carried out in AlMNC, the HAGB fraction decreased slightly to values of approximately 88% (Table 3), which indicates a weak response to this heat treatment.

### 3.3. Mechanical Properties

The Vickers microhardness of consolidate materials in as-extruded (HPE), as-solution-solid-treated (SST) and aged-hardening (T6) conditions, and percent difference between the hardness of T6 and SST samples ($\Delta H_{T6} = H_{T6} - H_{SST}$) are shown in Table 4.

**Table 4.** Vickers microhardness ($\mu HV_{100} \pm$ S.D.) of the AA6005A and AlMNCs milled powders in as-extruded (HPE), as-solid-solution-treated (SST) and aged-hardening (T6) conditions, and hardness percent difference between $H_{T6}$ and $H_{SST}$ ($\Delta H_{T6} = H_{T6} - H_{SST}$).

| Material | $H_{HPE}$ | $H_{SST}$ | $H_{T6}$ | $\Delta H_{T6}$ (%) |
|---|---|---|---|---|
| AA6005A | 62.92 ± 1.6 | 56.56 ± 1.4 | 75.48 ± 1.5 | 33.45 |
| AlMNC | 89.70 ± 2.9 | 82.13 ± 2.7 | 98.98 ± 2.8 | 20.51 |

A lower increase in hardness was observed after the T6 heat treatment in the extruded AlMNC-HEBM samples than in the extruded AA6005A-HEBM samples, (10.34% versus 19.96%).

This fact is confirmed by comparing the increase in the hardness of both types of samples after ageing heat treatment from the SST condition, as measured by the $\Delta H_{T6}$ values (20.51% versus 33.45%). This weak response to ageing heat treatment of AlMNCs was already reported in the literature for other Al-based composites [38,42]. According to Corrochano et al. [38], the decreased age-hardening ability of Al–Mg–Si/MoSi$_2$ composites, processed by P/M, is the main reason for the decrease in the matrix grain size. In the range studied, the hardness of composites in the SST condition follows a Hall–Petch mechanism, and their weak hardening response could be attributed to a higher density of grain boundaries that act as sinks for the solute atoms and vacancies, which are needed to form age-hardened precipitates, promoting a larger volume fraction of precipitate-free zones. The results obtained by Parvin et al. [45], in their study on the ageing behaviour of the SiC-reinforced Al6061 composite produced using HEBM, showed that milled composites exhibit no ageing hardenability. According to these works, composite materials not only have a high-dislocation density around ceramic particles but also enjoy ultrafine grain structures of submicron size, which are preferred precipitation sites. The age-hardening ability will decrease if the precipitation of the stable phases on the grain boundaries increases, because the stable phases make very little contribution to the hardening. They also reduce the solute atoms in the matrix. They concluded that the conversion of intermediate phases, which are located on the dislocations, into equilibrium phases and the growth of equilibrium phases becomes more active. Consequently, the precipitates could not strengthen composites. Knowles et al. [23] achieved different results by studying AA6061 reinforced with SiC nanoparticles. They found that 10 wt% SiC-containing composites exhibit a similar increase in hardness to that of unreinforced AA6061 after age-hardening, and 15 wt % SiC-containing composites exhibit less increase in hardness than unreinforced AA6061 after heat treatment, demonstrating that also in nanocomposites, there is a decrease in the ability to harden by ageing when the degree of particle reinforcement increases.

Table 4 and the DSC scans in Figure 5 indicate that the composites follow the expected general trend of higher solutionised hardness, shorter peak age time, and less age-hardening than the unreinforced alloys [38], and as expected, $\Delta H_{T6}$ decreases with the decreasing grain size.

The results of the tensile engineering stress–strain test are shown in Table 5. There is a clear trend for increasing the Young's modulus (E), yield stress (YS), and ultimate tensile stress (UTS) in both the as-extruded and aged conditions of AlMNCs-HEBM in comparison to as-extruded AA6005A-HEBM. The increase in E implies that there is a good load transfer occurring between the particles and matrix, by a strong bonding between Al alloy and n-TiC and with no brittle phase precipitating at the interface [23]. The AlMNCs samples in the as-extruded condition exhibit a relative greater increase in YS (approximately 40%) and UTS (approximately 20%) than in the T6 heat treatment condition (approximately 8.5%) with respect to the extruded composite, which confirms its weakness to the hardening by ageing. The strain to failure decreases remarkably with the n-TiC reinforcement, as would be expected when adding a brittle component, with a slight recuperation after the T6 heat treatment.

**Table 5.** Tensile testing results of the AA6005A-HEBM and AlMNC-HEBM. Young's modulus (E), yield stress (YS, 0.2% offset), ultimate tensile strength (UTS), strain to failure ($\varepsilon_f$), and average dimple size in as-extruded (HPE) and aged-hardening (T6) conditions.

| Materials | Values of Tensile Tests | | | |
|---|---|---|---|---|
| | E (GPa) | YS (MPa) | UTS (MPa) | $\varepsilon_f$ (%) |
| AA6005A (HPE) | 63 | 134.5 | 237.50 | 12.75 |
| AA6005A (HPE+T6) | 64 | 134.67 | 255.67 | 11.80 |
| AlMNC (HPE) | 70 | 188.00 | 306.33 | 5.00 |
| AlMNC (HPE+T6) | 73 | 204.00 | 332.33 | 7.50 |

## 4. Conclusions

A composite with a fine-grained AA6005A matrix and 3 vol% nanoparticles of TiC was produced by hot extrusion and T6 heat treatment of high-pressure gas-atomised and mechanically milled powders.

The influence of grain refinement on the microstructure and mechanical properties of the composite was investigated. The most important findings can be summarised as follows.

(1)　High-energy ball milling (HEBM) significantly refined the grain structure of the AA6005A matrix alloy, and a grain structure consisting of highly misoriented and approximately equiaxial grains of different grain sizes was formed.

(2)　The nanocomposite showed remarkably higher hardness, Young´s modulus, yield, and ultimate strengths at room temperature than the extruded profiles of mechanically milled unreinforced Al alloy powders.

(3)　The consolidated nanocomposite showed a moderate response to ageing heat treatment as demonstrated by microstructural analyses and mechanical tests. In any case, the composites produced in this research, combining the use of nanoparticles TiC and T6 heat treatment, display high strength and hardness although with a significant drop in ductility, as would be expected when adding a brittle component, and with a slight recuperation after the T6 heat treatment.

**Author Contributions:** Conceptualization, M.C.; Funding acquisition, G.P. and P.R.; Investigation, I.F.; Methodology, P.M. and G.P.; Supervision, M.C.; Validation, P.M. and P.R.; Writing—original draft, I.F. and P.M.; Writing—review & editing, M.C. All authors have read and agreed to the published version of the manuscript.

**Funding:** This research was funded by Ministry of Economy and Competitiveness of Spain grant number MAT2013-48166-C3.

**Conflicts of Interest:** The authors declare no conflict of interest.

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
