# Peer review of "Microstructure and Mechanical Properties of an Extruded 6005A Al Alloy Composite Reinforced with TiC Nanosized Particles and Strengthened by Precipitation Hardening"

_metals, doi:10.3390/met10081050_

Round 1

Reviewer 1 Report

It is a nice work. However a few points need to be improved prior to acceptance:

1- It is very difficult to understand the work due to language problem. The text need to edited by a professional.

2- Figures 2-4 are FESEM and not TEM micrographs. This section about sample preparation has to be recast- very confusing.

3- In Fig. 2, the average size of the individual particles is about 150-200 nm. To coarse to be called nano particles 

4- AlMMNC-HEBM-HPE too long code

5- in Fig 3 the C carbon appears in a long particle that has no Ti, why?

6- In Fig. 4 white particles appear on the surface of an Fe-based intermetrallic. What are these particles.

7- The DSC runs are good but peaks need to be well idendified in terms of the precipitated phases. Not long expressions.

Author Response

First of all, we want to thank the Referees and the Editor for their careful review of our paper, and for the comments, corrections and suggestions that ensued. A detail revision of the paper has been carried out to take most of them into account. And in the process, we believe the paper has been significantly improved, and we hope that you will find it for publication.

Review 1

It is a nice work. However a few points need to be improved prior to acceptance:

  • It is very difficult to understand the work due to language problem. The text need to edited by a professional.

Sorry about the difficulty. We have sent to a professional for editing. The corrections are include in the paper

  • Figures 2-4 are FESEM and not TEM micrographs. This section about sample preparation has to be recast- very confusing.

Sorry about the confusion, experimental procedure was no properly explain this part. The new paragraph -lines 183-186- is correct.

The Fig 2-4 are STEM. STEM is similar to TEM, it is necessary thin sample and we have used the field emission transmission electron microscopy. The TEM parallel electron beam are focused perpendicular to the same plane, in STEM the beam is focused at a large angle and is converged into a focal point. We try to analyze small particles and we could not see well by SEM.

  • In Fig. 2, the average size of the individual particles is about 150-200 nm. To coarse to be called nano particles 

We agree, but I think in the description of the Figure 2, we could not find the word nano associates to the isolated intermetallic compound from the alloy. Anyway we have change in the line 274 the reference of nano. The n-TiC is well distributed but we can fine some agglomerations in clusters (150 nm-200 nm)

  • AlMMNC-HEBM-HPE too long code

We agree, we have change all the codes in the all paper. AlMNC in the new AlMMN and we have tried to explain without using the long codes.

5- in Fig 3 the C carbon appears in a long particle that has no Ti, why?

Ti and C are always together. It is a lack or bad definition of C in all the particles, that is because the C signal is difficult to get with the EDS detector compared with the metals. We have remarked the interest área in the Figure

6- in Fig. 4 white particles appear on the surface of a Fe-based intermetallic. What are these particles?

They are the Al-Fe-Mn-Si described in the base alloy. They are in the AA6005 alloy line 254. We have indicated in the line 282.

7.-The DSC runs are good but peaks need to be well identified in terms of the precipitated phases. Not long expressions.

We have changed the long expressions and in the text is included the peaks description (lines 210-228) we have included in the Figure the peaks symbols (I, II, III)

Reviewer 2 Report

This is an interesting article. In this paper, the authors report a systematic study on the effect of TiC nanoparticles addition on the ageing behavior of an extruded AA6005A alloy. The nanocomposites were produced by a process consisting of high-energy ball milling and hot extrusion after uniaxial pressing. All the microstructure characterization and mechanical tests were carefully conducted and the results were presented clearly. The conclusion is well supported by the experimental results. I recommend it for publication in the journal.

Author Response

Thanks for your nice comments. I have improved the paper.

Reviewer 3 Report

Dear Sirs,

1) Row 109 and other places: "aluminium metal matrix nanocomposite".

Aluminium is metal. It is necessary to use (as n reference [3]) "aluminum matrix nanocomposite"

2) Authors use 3%vol TiC. How they calculate this volume fraction?

3) During ball milling: it is necessary to point mass of composites from 1 milling container.

4) Milling time: do you calculate full time (10 h) or treated time (without time of stops)

5) Rows 137-150: it is not possible to understand procedure of extrusion. Size of cylinder is absent. Is it possible to produce plate 40x3x1000 mm from cylinder?

6) Row 278:  "UM-...." is absent in picture.

7) May be good improvement of properties (Table 5) not from 3% TiC only, but also from oxides? You wrote about oxides.

Author Response

First of all, we want to thank the Referees and the Editor for their careful review of our paper, and for the comments, corrections and suggestions that ensued. A detail revision of the paper has been carried out to take most of them into account. And in the process, we believe the paper has been significantly improved, and we hope that you will find it for publication.

Review 3

It is a nice work. However a few points need to be improved prior to acceptance:

1) Row 109 and other places: "aluminium metal matrix nanocomposite".

Aluminium is metal. It is necessary to use (as n reference [3]) "aluminum matrix nanocomposite"

Thanks for your suggestion. We have changed in the whole paper.

2) Authors use 3%vol TiC. How they calculate this volume fraction?

There are many ways in the bibliography to express the % in the reinforcement. At the beginning of the research, during the bibliographic revision we found many papers, especially with nanoparticles, they used vol% instead to wt%. So we have used vol% in order to have a reference.

%Vol=100 x (Vol reinforcement (TiC))/Vol Total (6005+PCA+TiC)

We weight the quantity of nano TiC to have a specific weight percentage. We convert the weight percentage to volume percentage using the mass and density of all components of the composite: alloy, reinforcement and PCA

 3) During ball milling: it is necessary to point mass of composites from 1 milling container.

The whole mass put into the container is weighed before starting the process. You have to weight the whole mass (alloy, reinforcement and PCA) to create the composite and then to put all the components into the ball milling. You close the ball milling and starts the process. We have mill in several batches with the same percentage until we have the HPBM powder necessary the extrusion. Then we do not understand why we need to point mass of composite from 1 milling container.

With the powder we filled a cylindrical container of Al (40 mm diameter and 100 mm length).The percentage of TiC is 3 vol , because it is used from the beginning.

4) Milling time: do you calculate full time (10 h) or treated time (without time of stops)

10h means that 10h of effective milling time. The stops are only for cool down the composite and it is not considered on the full milling time. We have pointed out line 139.

5) Rows 137-150: it is not possible to understand procedure of extrusion. Size of cylinder is absent. Is it possible to produce plate 40x3x1000 mm from cylinder?

We agree, we have to include the cylinder container size (line 145). Then we have changed the full paragraph (line 141-158)

The cylinder container is heated and then press in the extrusion machine. The die has a rectangle section, then SF

6) Row 278:  "UM-...." is absent in picture.

Thanks a lot, we have change in the Figure 5.

7) May be good improvement of properties (Table 5) not from 3% TiC only, but also from oxides? You wrote about oxides.

Thanks for your comment. We wrote about oxides, but they are in all HPBM samples not only TiC. Then it is not consider the oxides in the improvement in the strength between AA6005 and AlMNC both in an extruded condition.

Round 2

Reviewer 1 Report

The authors did the requested modifications.